# Time separating spatial memories does not influence their integration in humans

Xiaoping Fang[1,2☉], Benjamin Alsbury-Nealy[1☉], Ying Wang[3,4], Paul W. Frankland[1,3,4,5,6], Sheena A. Josselyn[1,3,4,5], Margaret L. Schlichting[1☉], Katherine D. Duncan[1☉]*

1 Department of Psychology, University of Toronto, Toronto, Canada, 2 School of Psychology, Beijing Language and Culture University, Beijing, China, 3 Program in Neurosciences and Mental Health, The Hospital for Sick Children, Toronto, Canada, 4 Department of Physiology, University of Toronto, Toronto, Canada, 5 Institute of Medical Science, University of Toronto, Toronto, Ontario, Canada, 6 Child & Brain Development Program, Canadian Institute for Advanced Research (CIFAR), Toronto, Canada

☉ These authors contributed equally to this work.
* katherine.duncan@utoronto.ca

**Data Availability Statement:** The data in the study are available from the Open Science Framework (https://osf.io/se8t7).

## Abstract

Humans can navigate through similar environments—like grocery stores—by integrating across their memories to extract commonalities or by differentiating between each to find idiosyncratic locations. Here, we investigate one factor that might impact whether two related spatial memories are integrated or differentiated: Namely, the temporal delay between experiences. Rodents have been shown to integrate memories more often when they are formed within 6 hours of each other. To test if this effect influences how humans spontaneously integrate spatial memories, we had 131 participants search for rewards in two similar virtual environments. We separated these learning experiences by either 30 minutes, 3 hours, or 27 hours. Memory integration was assessed three days later. Participants were able to integrate and simultaneously differentiate related memories across experiences. However, neither memory integration nor differentiation was modulated by temporal delay, in contrast to previous work. We further showed that both the levels of initial memory reactivation during the second experience and memory generalization to novel environments were comparable across conditions. Moreover, perseveration toward the initial reward locations during the second experience was related positively to integration and negatively to differentiation—but again, these associations did not vary by delay. Our findings identify important boundary conditions on the translation of rodent memory mechanisms to humans, motivating more research to characterize how even fundamental memory mechanisms are conserved and diverge across species.

## Introduction

Humans sometimes leverage their prior knowledge to flexibly navigate even new spatial environments. For example, frequent travelers may know to search for fruit at the front of supermarkets when looking for a quick breakfast—knowledge gained by integrating across many grocery runs to different stores. Having these sorts of regularities stored in memory may help people more efficiently locate important places in a new environment, particularly

**Funding:** This work was supported by a Canadian Institutes of Health Research (CIHR) Project Grant (PJT-159492) to KDD, MLS, PWF, and SAJ; Natural Sciences and Engineering Research Council of Canada (NSERC) Discovery Grants (RGPIN-04933-2018 to MLS; RGPIN-05582-2016 to KDD); Canada Foundation for Innovation JELF to MLS and KDD; and Ontario Research Funds (36876 to MLS; 34479 to KDD).

**Competing interests:** The authors have declared that no competing interests exist.

when that new environment bears some resemblance to a previously experienced one. But, on the other hand, a seasoned traveller must also differentiate apparently related environments in their memory to, for example, locate their preferred coffee spot in similar looking airport terminals. Here, we investigate one factor that might impact whether two related spatial memories are integrated or differentiated: Namely, the temporal delay between experiences.

Past work in rodents has shown that two memories formed close together in time are more likely to become linked than those separated by long delays [1–4]. For example, one study [4] showed that extinguishing the response to one fear-conditioned stimulus can transfer to another when the initial learning was separated by a 6- but not a 24-hour (h) delay. Such generalization (or false memory of fear for a never-conditioned stimulus, depending on your perspective) is emblematic of a type of memory integration in which experiences become fused into one memory representation. Hereafter, we refer to this fusing as "integration." Indeed, this transfer of extinction was underpinned by the greater co-allocation of the memories to overlapping populations of neurons in the amygdala at shorter delays [4]. Other work—using contextual fear conditioning—reported conceptually similar findings in the hippocampus comparing 5h versus 24h [1] or 7 day (d) delays [1, 3]. Mechanistically, the bias to integrate memories formed close in time is governed by two key principles of memory allocation (for reviews see [5–9]): First, that neurons which happen to be more excitable during an experience are more likely to be allocated to the resulting memory trace [10–14]; and second, that neuronal excitability is persistently elevated for hours after a learning experience [15–17]—triggered by activation of the transcription factor CREB (cyclic adenosine monophosphate response element-binding protein) [11, 13, 18–24] and delayed expression of the immune receptor C-C chemokine type 5 (CCR5) [1]. Thus, events that follow shortly after a learning experience are more likely to be allocated to an overlapping neuronal population [11–14, 25–30]. What we do not yet understand is whether the same mechanisms also govern the integration of spatial memories in humans.

Research with human participants has provided some preliminary evidence for the conservation of this phenomenon across species, in that non-spatial memories appear to be more interconnected when formed at a short as compared with a long delay. One example study [31] used a pair learning task to show that people were better able to make novel inferences across multiple memories formed on the same (30 minute [m] delay) versus different (24h delay) days [31]. However, this enhanced integration was not related to the heightened activation of the initial memory during related learning (assessed with multivariate fMRI techniques), as would be expected from the rodent literature. Rather, it was attributable to participants' spontaneously adopting an integrative encoding strategy more often in the short than long delay condition [31]. Further pointing to a strategic mechanism, short and long delay conditions also evoked differences in prefrontal (but not hippocampal) engagement during the inference decision itself. Other work [32] used fear conditioning to ask if fear associations more readily enhanced the memory for neutral lists learned close (5h) as compared to far (7d) apart in time. Consistent with the spreading of fear associations, lists were more resistant to forgetting in the short compared to long delay conditions, but only under select situations (i.e., when tested in a novel but not familiar context; see also [33]). Moreover, final memory performance was not higher in the short compared to long delay conditions, suggesting that initial memory differences may have also contributed to the forgetting effect. Therefore, open questions remain as to why the results were restricted to forgetting and only observed in some conditions—both of which are challenging to reconcile with a co-allocation mechanism. Thus, in sum, evidence for time-dependent memory co-allocation in humans appears both more mechanistically ambiguous and less robust than that observed in rodents. But, with vastly

different paradigms used across species, it is unclear whether these discrepant findings are attributable to species or task differences.

Further complicating matters, human spatial memories have been characterized as differentiated, even when acquired at very short delays. For example, two experiments using analogous paradigms first trained participants to place the same objects in target locations within two visually similar arenas [34, 35]. Germain to our question, participants experienced both environments within the same session without significant intervening delays. Later, they placed objects in morphed arenas that shared elements of both training environments during fMRI. If they had integrated learning into a single mental map, one would expect them to place objects in "morphed" locations, situated between directly trained ones. However, participants only placed objects in trained locations, and their patterns of hippocampal activity mirrored that observed in the most similar trained arena [34, 35]. This suggests that people spontaneously form highly differentiated hippocampal representations of environments–even those experienced close in time–such that later navigation reflects the attraction of hippocampal activity to one representation. However, aspects of the paradigms may have fostered differentiation. For one, participants traveled from one arena to another over a bridge [35]. In rodents, a similar manipulation (adding a corridor connecting arenas during training) yields more differentiated hippocampal representations relative to when each arena is placed within the same global context [36]. More pragmatically, given that the targets were positioned in discrete locations, it is unclear how their locations could be integrated into *intermediate* locations. Perhaps probabilistic distributions—like fruit tending to be toward the front of supermarkets—are more conducive to integration. Given that these paradigms may have pushed the system towards differentiation, it remains an open possibility that spatial memories acquired close in time could be integrated under more neutral learning conditions.

Here, we aim to develop a more neutral assessment of spatial memory; one which could allow integration in both human and rodent memory, and be sensitive to the generalization and blurring thought to result from fused memory representations. In contrast to past work [34, 35]—and to allow integration—we created reward distributions within arenas that were positioned within the same global environment. Our approach was also influenced by three paradigmatic considerations for species translation. First, while relevant human studies tend to test memory for dozens of briefly presented discrete images, rodent studies tend to test a small number of more enduring experiences. Second, while human memory integration tends to be studied with inference judgements [37, 38], in which correct responses can be supported by complex strategies during learning or retrieval, the rodent co-allocation work reviewed above assessed integration through measures like extinction and transfer, in which there is no optimal (strategically supported) response. Lastly, while human memory integration is typically studied using appetitive or unmotivated learning tasks, the impact of time on memory co-allocation in rodents has been studied exclusively in fear conditioning contexts. Because such tasks rely on freezing as the sole behavioural measure, they do not offer a nuanced picture as to how integrated memories may also support complex, instrumental behaviours. Thus, with the ultimate goal of enabling the direct cross-species comparison needed to deepen our understanding of memory integration, we devised a task that could assess how reward-seeking, appetitive behaviours spontaneously transfer between two spatial environments.

Here, we report the human component of this endeavor. We had participants learn to locate rewards in two virtual environments that differed in local features (arena shape: circle or square) but were situated in the same position within the same global context (Fig 1A). The global context provided a relatively homogenous backdrop that was designed to mimic a rodent testing room as closely as possible; there were only four simple shapes positioned at the cardinal directions to enable orientation within the environment. Participants first completed

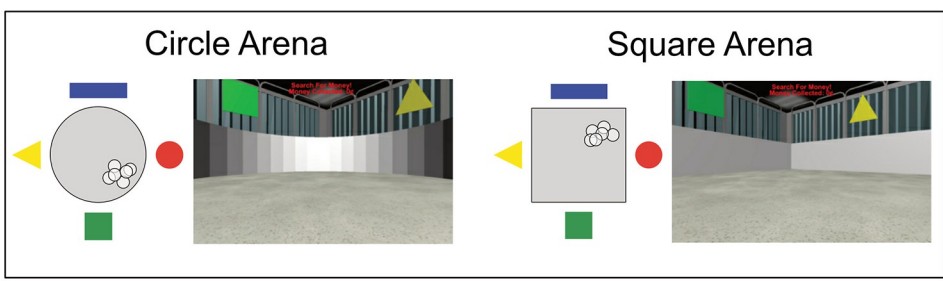

**Fig 1. Navigation task overview and learning performance.** (**A**) 3D environment stimuli. Circle (left) and square (right) arenas shown during the learning phase. For each shape, an aerial view depicting the colourful shapes that appeared at the four cardinal directions as well as the reward points (white circles) within the arena (light grey) is shown on the left. All participants experienced the same six reward points in each arena; their locations and sizes are to scale. First-person perspectives during navigation are shown on the right. (**B**) Experiment structure. Participants first completed a learning phase (left) split across two sessions (Session 1, dark grey; Session 2, light grey), which were separated by a 30min (orange), 3h (pink), or 27h (indigo) delay. In each session, participants learned to find rewards (white circles) in either the circle or square arena; assignment of arena shape to session was counterbalanced across participants. Only one such assignment is depicted here for simplicity. Participants then completed the test phase (right) three days later, which included first a Direct test for the reward locations in the two trained arenas (top), and then a Transfer test that also incorporated three novel morphs (bottom; 6-, 8-, and 12-sided, depicted as middle three shapes along continuum). There was no feedback in the tests, represented by a lack of white circles in these diagrams. (**C**) Learning performance as estimated by the proportion of time spent in the Target (rewarded) Zone (see Fig 2A) at the end of Session 1 (left) and Session 2 (right). Results depict performance on the last two no-feedback trials.

Participants in the 3h delay condition performed best overall, spending more time in the target zone than the other groups in Session 1 and showing a trend of more time than the 30min group in Session 2. $*p<0.05$, $\sim p < .10$.

a learning phase (Fig 1B, left), in which they searched for rewards hidden in each arena across learning sessions separated by delays of either 30min (preregistration https://osf.io/gnxpt), 3h, or 27h (https://osf.io/49dtx). Three days after the second session, they completed a memory test (Fig 1B, right), in which they searched for rewards in both trained environments (Direct test) as well as in novel morph environments that contained enclosure shapes that were in between the trained circle and square (Transfer test). We reasoned that integration across arenas would result in the merging of cognitive maps for the two spatial environments, which at its most extreme would mean participants search in a unified region, regardless of the arena in which they were placed. Alternatively, differentiated memories would be evidenced by search trajectories restricted to only the rewarded portion of the arena being tested. We predicted that greater memory integration would be seen when there was a short (30min, 3h) as compared with a long (27h) delay between learning experiences.

## Materials & methods

### Participants

One hundred and thirty-one young adults (23.8 ± 4.3 years old; 81 females, 50 males) recruited through the local community, participant databases, and online platforms participated in the study (n = 42–45 per delay condition; see Table S1 in S2 File). Individuals were eligible for the study only if they reported being fluent English speakers, having normal or corrected-to-normal vision, not having any history of neurological disorders or mental illness, and not being on psychoactive medication. Participants provided informed consent prior to the experiment and were compensated at a rate of C$10 per hour plus a bonus according to performance (average bonus: C$19.50 ± 9.30). An additional 116 participants were recruited from the same sources but did not complete the study because they either (1) failed to reach one of our three performance criteria (more details below; failing practice: n = 50; failing Session 1: n = 55; failing Session 2: n = 5) or (2) did not return for Session 2 (n = 6). Another two participants (both in the 27h condition) completed all three sessions but were excluded from the final analysis due to poor computer performance that resulted in low temporal resolution data (< 10 Hz). The protocol was approved by the University of Toronto Research Ethics Board, and all participants gave written informed consent.

### Sample size determination

Data collection was carried out in two "waves" (Wave 1: July 21—December 23, 2020; Wave 2: February 22—June 5, 2021), and online due to the COVID-19 pandemic. In Wave 1, we compared memory at 3h and 27h delays. We found similar levels of memory integration and differentiation across delays and, therefore, added a third delay of 30min—the shortest logistically feasible delay—in Wave 2 to assess the possibility that memory integration is only heightened at very short delays in humans. We separately preregistered the two waves (Wave 1: https://osf.io/49dtx; Wave 2: https://osf.io/gnxpt). We set the target sample size and stopping rules in Wave 1 using a sequential analysis approach [39] (discussed in *Supporting Methods in S1 File, Preregistered sequential analyses and stopping procedures*). We predetermined Wave 2 sample size to match the group sizes from Wave 1.

## Study overview

In the main memory experiment, participants learned about reward distributions in two distinct arenas that shared a global context. The two arenas were presented to participants in different experimental sessions (Sessions 1 and 2) separated by either 30min, 3h, or 27h. Reward distributions across the two arenas were positioned close to one another with respect to the global context (i.e., in neighbouring quadrants), which enabled either integration into a single mental map or differentiation into two separate ones. In all conditions, the final memory test occurred three days after Session 2 (Fig 1B). We also collected several individual difference measures at the end of the study (discussed in *Supporting Methods, Supplementary Measures in S1 File*).

## Virtual environment stimuli

We created virtual environments using OpenMaze [40] (https://openmaze.duncanlab.org/), an open-source toolbox for Unity Software (Unity Technologies). Most of the time, participants saw the 3D environments from a first-person perspective; during some task phases, they were presented with aerial views. In all cases, they used arrow keys to navigate.

The global context for the main experiment mirrored a rodent testing room: It was relatively homogenous, with hospital curtains and only a simple coloured shape positioned at each of the four cardinal directions to enable orientation. The local arena was positioned in the center of the global context, and its shape was manipulated across phases of the task. Training occurred in a 4-sided "square" arena or a 40-sided arena that approximated a circle (hereafter, "circle" for brevity), but some testing occurred in "morphs"—regular polygons with intermediate numbers of sides. Throughout, we use the term "environment" to refer to the combination of the arena and global context.

**Norming study for morph arena development and selection.** To select morph shapes, we conducted an online norming study using Inquisit 5 Software on Amazon's Mechanical Turk. Participants (n = 40) were sequentially presented with a pair of environment screenshots from a 1st person perspective. Half of the pairs were identical; the other half only differed in enclosure shape (4–12 sides). Participants were asked to indicate if the images were identical or different after seeing both presented for 1s each, separated by a 1s coloured mask. We then identified the optimal spacing of morphs such that their perceptual discriminability was roughly equated. This procedure led to the selection of morphs with 6, 8, and 12 sides.

## Procedures

Participants first practiced an abbreviated version of the main experiment's learning phase described below within a distinct virtual environment (described in *Supporting Methods in S1 File*, *Practice task procedures*, and depicted in Fig S1B in S2 File). Those who exceeded our performance criterion for the practice continued to the main experiment, which had three phases: Learning (split across two sessions), Direct test, and Transfer test. During learning, participants searched for rewards hidden in either a square or a circle arena. Reward quadrants and specific locations were fixed across participants and were exactly as shown in Fig 1, such that (for instance) the rewards in the circle arena always appeared in the bottom right quadrant. Only one arena shape was presented per learning session, with their order counterbalanced across participants (see Fig S1A in S2 File for a schematic depiction). During the Direct test, the circle and square arenas were presented again to assess memory for the rewarded locations. Then, during the Transfer test, we additionally assessed the generalization of learning to new "morph" arenas. To assess the generalization of memories to novel arenas, we included a

Transfer test in which participants experienced the three new arena shapes, or "morphs," in addition to the trained circle and square arenas.

**Learning tasks.** *Environment familiarization.* Participants were instructed to freely explore the 3D environment. Thirty seconds later, they were presented with an aerial view of the environment and were asked to navigate to the last location they had occupied. Their current position was indicated on the screen with a small triangle, which always was first placed in the center of the arena (Fig S1A in S2 File). The participant then returned to the 3D environment and repeated the exploration and location reporting procedure five times. For all 3D environment components, participants' starting locations within the arena and initial head directions were fully random to facilitate an allocentric representation of the space.

*Search familiarization.* Participants were cued to search for coins (equivalent to half a cent) hidden in the now-familiar arena. The invisible coins were collected by walking through their location. A sound played to signal the collection of each coin, and the total number collected was displayed on the top of the screen. The coins were densely packed within a small circular area, which we hereafter term a "point." Coins were regenerated immediately after participants left their location, so that an unlimited number could be collected, provided that participants kept moving. Participants had unlimited time to collect 20 coins, and the phase terminated once participants achieved the goal.

*Reward searching.* Like in Search Familiarization, participants searched for invisible coins. On each trial, the coins were densely packed within one of six points drawn from the same Gaussian distribution centered in a quadrant of the arena (Fig 1A); the first trial used the same point as was used in *Search Familiarization*. Participants were not told that the reward locations would change across trials, to mimic mice experiments in which such instruction is impossible. However, they did experience changing reward locations during the preceding practice task, and, as such, likely anticipated this task element.

In the Reward Searching phase, 30 feedback trials were interspersed with seven no-feedback trials, with no-feedback trials becoming less frequent as training progressed. Before each trial, participants were cued to the trial type (feedback or no-feedback) and then given 1m to collect as many coins as possible. Participants had to collect at least 20 coins per trial to add the earnings to their bonus pay. In the feedback trials, participants heard coin sounds, and the number of coins collected in the current trial was displayed. In the no-feedback trials, participants did not receive any of this feedback. Participants were rewarded at each of the six points in three consecutive feedback trials before the reward location moved to the next point (Fig S1A in S2 File). The same sequence of points was then used for two consecutive feedback trials each, resulting in five feedback trials per point in this phase.

To ensure participants successfully learned about the reward locations in each arena, we calculated the average time spent in the reward zone during the last four no-feedback trials; only those participants who spent at least 30% of their time in this zone moved on to the next session. Here, a reward zone was defined as the largest inscribed circle in a quadrant where rewards are located.

**Direct and transfer tests.** Participants' memory was tested first in each learned arena (Direct test) and then in three novel morph arenas to assess generalization (Transfer test). On each test trial, participants had one minute to collect as many coins as they could without feedback. In the Direct test, circle and square arenas were each tested three times in alternation. In the Transfer test, participants were instructed to use what they learned previously about the environments to guide their search for rewards in each arena (trained circle and square plus three novel morphs). Note that we included the directly learned arenas during this phase so that we could examine search patterns across five shape points, necessary for assessing attractor-like representations with sigmoid models. We reasoned that it would be

inappropriate to use performance on the Direct test to estimate these endpoints because performance could change as a function of time and exposure to interfering arena shapes. Arenas were randomly intermixed, with each arena tested twice (once per block for two blocks).

## Data analysis

Full data analysis details are provided in the *Supporting Methods in S1 File*. Briefly, we recorded participants' movements as they searched the different arenas, then downsampled and filtered the data to eliminate uninformative time points (i.e., those very early in the trial or when no movement occurred; *Supporting Methods in S1 File, Data acquisition and preprocessing*). Our primary dependent measure was the proportion of time in a given trial spent in each of four key "Zones" (time-in-zone; TIZ): Target Zone, rewarded in the present arena; Alternate Zone ("Alt" for brevity), rewarded in the alternate arena; Adjacent Zone ("Adj"), neighboring the Target Zone but not itself rewarded; and Control Zone ("Ctrl"), opposite the Target Zone and also never rewarded. Note that the zones rewarded in each arena were fixed across participants (e.g., the bottom right quadrant was always the target zone of the circle arena).

For each participant and environment, we quantified initial learning performance as the $TIZ_{Target}$ at the end of learning (considering only the last two no-feedback trials). As we found an unexpected statistical trend for differences in initial learning across delay conditions (Fig 1C), in subsequent analyses, we deviate from our preregistered data analysis plan by including Session 1 learning as a covariate to ensure that any effects of delay are not attributable to group differences in initial learning. We also report the originally planned models (see S3 File); results remained unchanged.

We next used TIZ measurements to create indices of memory integration and differentiation based on Direct test behaviour. We reasoned that differentiation would be reflected in participants' greater tendency to search the Target as compared with the Alternate Zone, as this preference requires disambiguating between the present and similar, related arena in memory. Accordingly, we computed memory differentiation as $TIZ_{Target}$—$TIZ_{Alt}$. We preregistered this as our primary confirmatory approach with the logic that values close to 0 would reflect integration. But to more directly probe integration, we reasoned that memory integration would be reflected in participants' greater tendency to search the Alternate as compared with the Adjacent Zone: equated in their spatial proximity to the Target Zone, these zones only differed in their history of reward in the related arena. Therefore, in follow-up exploratory analyses, we computed memory integration as $TIZ_{Alt}$—$TIZ_{Adj}$, with a positive value indicating greater evidence for the merging of mental maps. Note that these metrics are related, in that more time in the Alternate Zone is interpreted as reflecting both higher memory integration and lower differentiation, but that both scores can be high when participants spend an intermediate amount of time in the Alternate Zone. We used a similar logic to quantify Session 1 reactivation and perseveration during Session 2 learning in other exploratory analyses (*Supporting Methods in S1 File*).

We additionally characterized each participant's Transfer test performance in a preregistered exploratory analysis by assessing whether their search paths within novel morphs more closely reflect differentiation into two attractor states or more gradual shifts within an integrated reward zone (*Supporting Methods in S1 File, Measuring memory generalization during the Transfer test*). For each arena in the Transfer test (two trained arenas and three morphs), we calculated a difference score reflecting the degree to which search behaviours mirrored the circle versus square arena reward locations from training ($TIZ_{Target\,|\,Circle}$—$TIZ_{Target\,|\,Square}$). Spending the majority of one's time in the "correct" zone would therefore yield a positive value if the tested arena were a circle, and a negative value if it were a square. We then followed

the approach used in [35] to fit individual participant's curves to both linear and sigmoid models and compared the models' fits to assess condition-related and individual differences. Importantly, because the behaviour of participants with poor memory for the reward locations would also likely be better-described by a linear model—inflating our estimate of integration —we required participants to show above-chance memory in at least one of the two arenas in the Direct test to be included in this analysis. In a follow-up exploratory analysis, we also applied the stricter threshold of showing memory for both arenas in the Direct test, to address the possibility that participants who only remember one arena may appear to have an integrated representation of both.

Statistical analysis was conducted in R (version 4.1.2). For our primary frequentist analyses we used a combination of mixed-effects and linear regression models depending on if we had multiple observations per participant; we additionally performed Bayesian analyses to assess evidence for the null when statistically non-significant condition differences were found with frequentist models.

## Results

### Initial learning performance

We assessed initial learning by focussing on the final two no-feedback trials during each of the two learning sessions (termed Sessions 1 and 2; Fig 1C). There was no effect of counterbalancing group (i.e., whether the circle or square arena was experienced first) on learning performance ($t(129) = -0.33$, $p = .740$), and as such, all data reported are collapsed across both orders. On average, participants spent 67.6% (95% confidence interval or CI = [64.4%, 70.9%]) of their time in the zone which contained rewards during Session 1 and 74.4% (95% CI = [71.2%, 77.6%]) during Session 2. This reflected stronger learning by the end of Session 2 compared with Session 1 ($\beta = 0.07$, SE = 0.02, $t = 3.53$, $p < .001$). Similar observations of transfer effects have been attributed to memory integration in related rodent work [3, 18], but we note that it could also reflect learning general skills required by the task, like using cues to navigate or developing strategies to find hidden rewards.

Delay condition was marginally related to participants' learning ($\chi^2 (4) = 8.64$, $p = .071$), with participants in the 3h condition generally outperforming those in the other two groups— a difference that was (surprisingly) present even in Session 1, before the delay manipulation had been introduced. Specifically, in Session 1, 3h participants spent significantly more time in the reward containing zone than did both the 30min ($\beta = 0.08$, SE = 0.04, $t = 2.09$, $p = .038$) and 27h ($\beta = 0.09$, SE = 0.04, $t = 2.19$, $p = .030$) groups and, in Session 2, showed a trend of more time than did the 30min group ($\beta = 0.08$, SE = 0.04, $t = 1.96$, $p = .052$); all other pairwise comparisons were not significant (i.e., there were no differences between the 30min group and the 27h group; $|t|<1.24$, $p>0.21$). While Session 2 differences could be related to our delay manipulation, their presence during Session 1 suggests that they may instead reflect naturally occurring variability across individuals. Accordingly, we included Session 1 learning as a covariate in the following Direct and Transfer test models. Our groups did not significantly differ in any measured attributes, including sex, age, self-reported spatial, navigational, time-of-day preference, or virtual navigation skills measured after the main experiment (see *Supporting Methods, Supplementary Measures in S1 File*, Table S1 and Fig S2 in S2 File), so these were not added as covariates.

Notably, improvements from Session 1 to 2 learning did not depend on delay ($\chi^2 (2) = 2.06$, $p = .357$), and were only statistically significant within the 27h group (27h: $\beta = 0.11$, SE = 0.03, $t = 3.14$, $p = .002$; 30min: $\beta = 0.05$, SE = 0.03, $t = 1.55$, $p = .125$; 3h: $\beta = 0.05$, SE = 0.03, $t = 1.40$, $p = .165$). This contrasts with some previous reports that rodents are more likely to transfer

learning across sessions when they are separated by shorter delays [3, 4]; we see no evidence for greater transfer (whether this be through memory integration or general task transfer) at shorter delays.

## Direct memory at a three-day delay

To assess memory integration and differentiation more precisely, we next turned to our primary metric: Direct test performance. Three days after completing the learning sessions, participants searched for rewards in both arenas. As detailed in the **Methods**, we derived Integration and Differentiation indices from the time spent in three key zones (Fig 2A displays zone definition and Fig 2B displays the proportion of time spent in each zone; see Table S2 in S2 File for comparisons of individual zones).

Turning first to integration ($TIZ_{Alt}$—$TIZ_{Adj}$), generally, we found that participants spent more time in the Alternate than in Adjacent Zones ($\beta = 0.14$, SE = 0.01, $z = 12.15$, $p < .001$), suggesting that some integration did occur. This integration was asymmetric: Specifically, participants showed more evidence of memory integration when they were in the first-learned arena, as indicated by a main effect of Session (Session 1 > Session 2; $\beta = 0.07$, SE = 0.02, $t = 3.68$, $p < .001$). Because this effect reflected a preference for (Alternate) Session 2 target locations, it could be consistent with either a recency bias or the asymmetric integration of Session 2 learning into Session 1 memories [41–43]. Consistent with the recency interpretation, participants spent more time in the second as compared to the first session's target locations when we collapsed across arena shape ($\beta = 0.13$, SE = 0.02, $t = 7.54$, $p < .001$). Note, though, that this *recency* bias partly reflects asymmetries in initial learning—not just the temporal proximity to the test—because it was correlated with superior Session 2 relative to Session 1 learning ($r_s$ (126) = 0.26, $p = .003$). Crucially, we found no evidence that temporal delay influenced memory integration ($\chi^2$ (2) = 1.10, $p = .578$; Fig 2C); in fact, there was Bayesian evidence in support of the null hypothesis (main effect: $BF_{01} = 1334.84$; 29 < pairwise $BF_{01}$ < 44). Taken together, the integration analyses therefore suggest that while initial spatial memories served as a scaffold into which memories for new reward locations were encoded, the degree to which this happened was delay-invariant.

Despite showing some evidence of memory integration, participants nonetheless remained able to differentiate between the two rewarded arenas: Overall, they spent more time in the Target than Alternate Zones ($\beta = 0.27$, SE = 0.01, $z = 23.59$, $p < .001$). As in the integration results, participants preferred searching in more recently learned locations, yielding greater differentiation scores when in Session 2 compared to Session 1 arenas (main effect of Session: $\beta = 0.22$, SE = 0.03, $t = 6.78$, $p < .001$). This asymmetry is also consistent with the recency bias described above. Most importantly, we also did not find any significant evidence for an effect of delay ($\chi^2$ (2) = 0.13, $p = .938$) on differentiation—and again, we in fact saw Bayesian evidence for the null (main effect $BF_{01} = 760.08$; 24 < pairwise $BF_{01}$ < 28).

Taken together, our investigation of both memory integration and differentiation metrics revealed that participants were able to disambiguate—yet simultaneously showed some crosstalk—between memories for the two arenas. Moreover, and contrary to our hypotheses, the extent of integration and differentiation was not influenced by the temporal delay between experiences. Rather, participants showed an overall bias toward the second-learned reward location.

## Transfer performance at a three-day delay

In the Transfer test, we asked participants to navigate novel morph arenas—which were similar but not identical to trained ones—with the reasoning that navigation patterns across

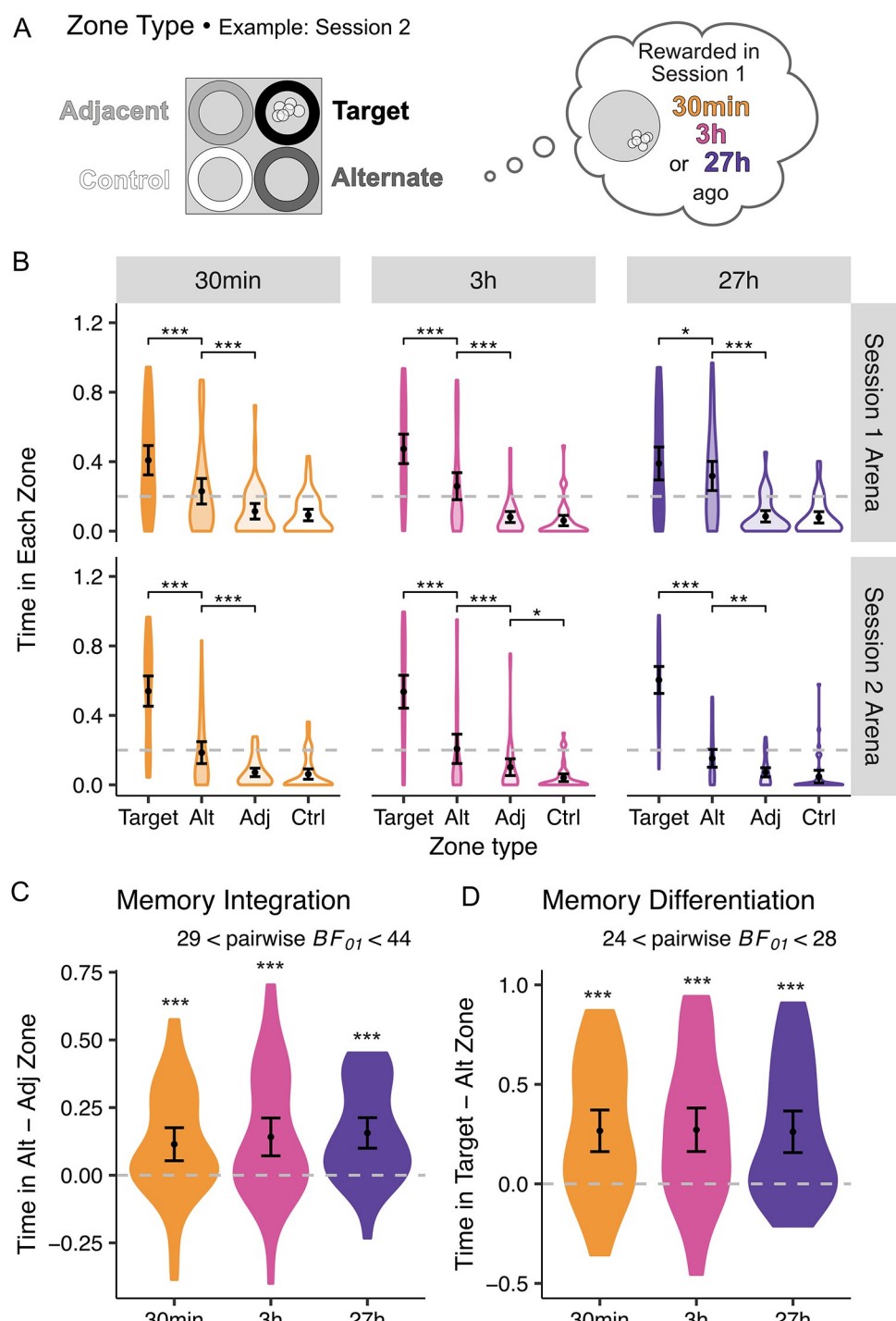

**Fig 2. Zone definition and Direct test results.** (**A**) Diagram showing Zone Type as it would be defined for one example arena (square) assigned to Session 2. Target Zone (darkest grey, top right quadrant) was where the reward was located in the current arena; the Alternate Zone ("Alt"; dark grey, bottom right) was in the quadrant rewarded in the alternate arena (corresponding circle arena from Session 1 included for reference); the Adjacent Zone ("Adj"; light grey, top left) was next to the Target Zone, but never rewarded; and the Control Zone ("Ctrl"; white, bottom left) was the farthest away from the Target Zone and never rewarded. Shade in circle shade maps onto the shades used in panel B graphs. (**B**) Reveals similar time in zone (TIZ) by zone type (shade) patterns for each delay (colour), stratified by session (row) of the Direct test. (**C**) and (**D**) show similar patterns of memory integration ($TIZ_{Alt}$—$TIZ_{Adj}$) and differentiation ($TIZ_{Target}$—$TIZ_{Alt}$) across delays (colour), respectively. See also Fig S4 in S2 File for data in panels **C** and **D** split by session. In all panels, *p<0.05, **p<0.01, ***p<0.005.

morphs could reveal subtle differences in memory structure. At the group level, participants significantly modulated their search behaviour according to arena shape ($\chi^2$ (12) = 267.21, $p <$ .001; Fig 3A), systematically spending more time in the zone previously rewarded in the most similar looking arena. However, this pattern was not impacted by temporal delay ($\chi^2$ (8) = 7.85, $p = .448$).

Having shown that search behaviour varied by morph shape, we next turn to our key measure of interest: Namely, whether individual participant's behaviour is best described by (1) a straight line, suggesting that learning is integrated across sessions to guide behaviour; or (2) a

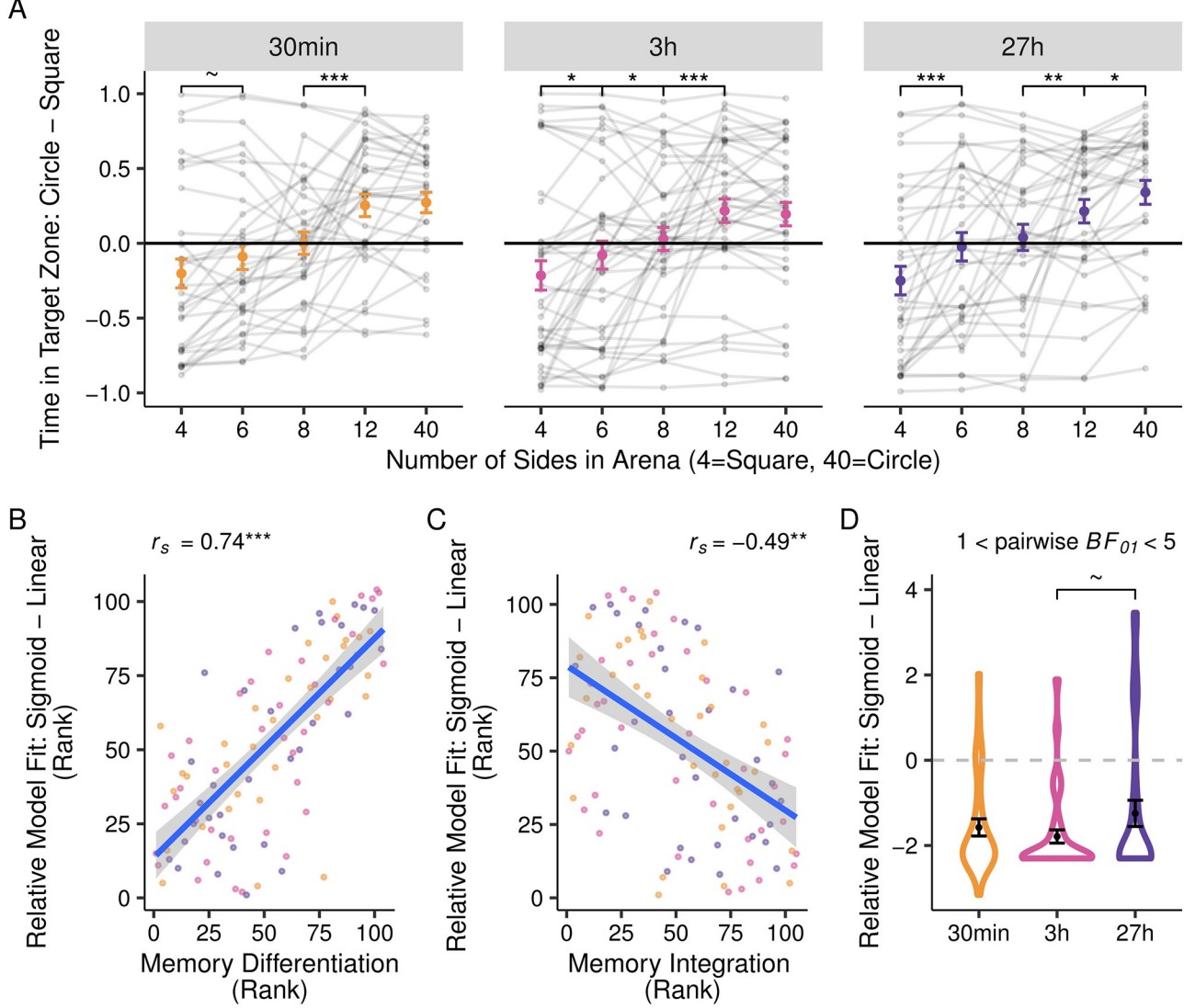

**Fig 3. Transfer test results.** (**A**) Relative time spent in circle vs. square reward zone locations (y-axis) as a function of arena shape (square → circle; x-axis). Individual participants are plotted in grey lines; group average and SEM are in colour. All groups tended to occupy locations rewarded in the most similar looking arena. (**B, C**) Model comparison of Transfer test behaviour (y-axis) are positively correlated with memory differentiation (Target—Alternate Zone time) and negatively correlated with memory integration (Alternate—Adjacent Zone time) in the Direct test (x-axes). Note that ranks are shown rather than values to mirror the Spearman correlation approach. (**D**) Relative fit of sigmoid vs. linear models (y-axis), by delay condition (x-axis). Positive values reflect superior fit of sigmoid over linear models; negative values the inverse. Model fit differences did not differ across delay condition. ~$p<0.10$, *$p<0.05$, **$p<0.01$, ***$p<.001$.

sigmoid, suggesting that the two reward zones act as separate attractor states, as would be expected under differentiation [35]. Overall, linear models described search patterns better than did sigmoidal models across the full group (taking into account model complexity; mean BIC values: Sigmoid = 0.394, Linear = -0.703; Wilcoxon signed-rank test, V = 4940, $p < .001$). However, we also saw variability in the degree to which participants' behaviour could be explained by each model. We validated these individual differences by relating them to variability in Direct test performance. Participants who showed a more sigmoidal pattern in the Transfer test (regardless of delay) also showed greater differences between Target and Alternate Zones in the Direct test ($r_s$ (102) = 0.74, $p < .001$; Fig 3B)—both patterns that we posited to reflect memory differentiation. Conversely, those participants who showed more sigmoidal search pattern in the Transfer test also showed less memory integration in the Direct test, or smaller differences between Alternate and Adjacent zones ($r_s$ (103) = -0.49, $p < .001$; Fig 3C).

Having validated our modeling approach, we next asked whether temporal delay influenced the relative fit of sigmoid and linear models in the Transfer test. This approach again revealed no effect of temporal delay ($F(2,99) = 1.68$, $p = .191$; main effect: $BF_{01} = 10.61$; 1 < pairwise $BF_{01} < 5$; Fig 3D). However, follow-up pairwise comparisons revealed one statistical trend, in which search paths in the 27h group were marginally less well fit by the linear model than the 3h group ($\beta = 0.59$, SE = 0.32, $t = 1.83$, $p = .070$). Yet we are hesitant to interpret this trend as there was no significant main effect and the data overwhelming show that the delay separating two spatial learning experiences does not have a measurable impact how humans integrated (or differentiated) the content of this learning.

We also included a follow-up analysis, to address the possibility that participants who only remembered one arena were inappropriately classified as having integrated the two arenas in their memory. When restricting the sample to the 78 participants who appeared to recall both arena's reward locations in the direct test, we again found that behaviour was better described by the linear model (mean BIC values: Sigmoid = 0.715, Linear = -0.223; Wilcoxon signed-rank test, V = 2433, $p < .001$) and was correlated with both differentiation and integration from the Direct test (sigmoid-differentiation correlation: $r_s$ (71) = 0.79, $p < .001$; sigmoid-integration correlation: $r_s$ (73) = -0.58, $p < .001$). However, under these more restrictive criteria, we observed a significant effect of temporal delay on the model fit ($F(2,70) = 4.70$, $p = .012$), such that behaviour in the 3h ($n = 28$) group was significantly more linear than it was in the 27h group ($n = 26$; $\beta = -1.22$, SE = 0.40, $t = 3.06$, $p = .003$; Fig S8 in S2 File). These results might support our hypothesized effect of temporal delay on memory integration, but only when memories are maintained well enough to reveal it. However, we caution against strongly interpreting these exploratory analyses in the context of the evidence for the null reported in our other analyses because they may reflect a survival bias. Specifically, this stricter criterion may have been particularly restrictive for the 27h group (whose Session 1 memory was tested at the longest delay), leaving only participants who had acquired particularly differentiated memories to begin with. Thus, this group could appear to have the most differentiated memories due to the subsampling rather than the delay itself.

## Might memory reactivation during Session 2 bridge long delays?

In light of this strong evidence that delay does not influence memory integration and differentiation in our paradigm, we performed exploratory analyses to understand why this might be. One possibility is that humans might be inclined to recall Session 1 learning during Session 2, even after a 27h delay. And, this memory reactivation could restart the same CREB-mediated excitability that normally would only facilitate integration at short delays. Indeed, reminding rodents of Session 1 learning shortly before Session 2 has been shown to remove delay's

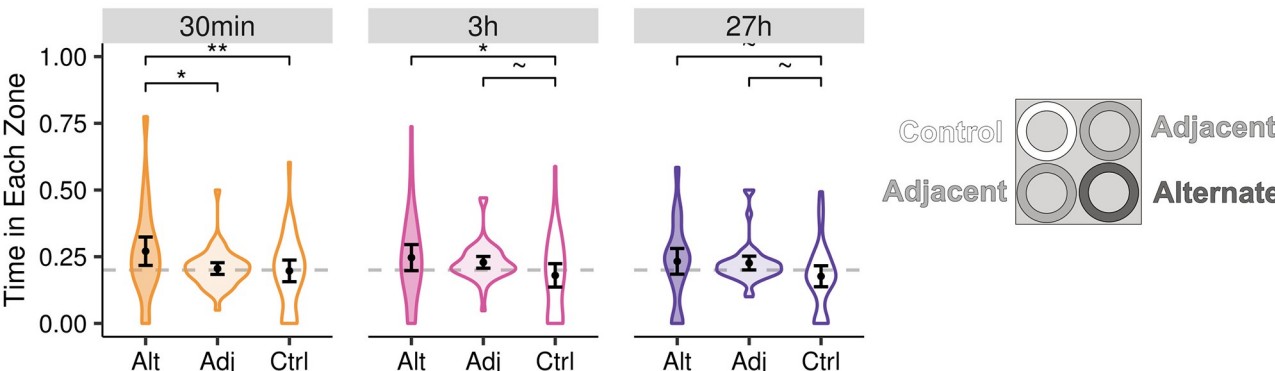

**Fig 4. Reactivation of Session 1 memories during initial Session 2 experience.** Participants showed overall reactivation of reward locations learned during Session 1, as evidenced by their spending more time in the Alternate as compared with Adjacent and Control Zones. ~$p$<0.10, *$p$<0.05, **$p$<0.01.

influence over integration, interpreted with this same reasoning [4]. We, therefore, estimated the degree to which participants seemed to spontaneously reactivate their Session 1 knowledge at the beginning of Session 2—importantly, before any rewards were delivered in the new Target Zone. Mirroring our memory integration metric, we compared the proportion of time participants spent during an initial exploration phase at the beginning of Session 2 in the Alternate (which was previously rewarded in Session 1) relative to the never-rewarded Adjacent and Control Zones. We then compared these results across delay conditions.

We found that overall, participants did indeed search more in the Alternate than the Control Zone ($\beta$ = 0.07, $SE$ = 0.02, $t$ = 4.05, $p$ < .001) and marginally longer than the Adjacent Zone ($\beta$ = 0.03, $SE$ = 0.02, $t$ = 1.84, $p$ = .067), consistent with them remembering prior Session 1 reward locations during the new Session 2 experience (Fig 4). However, memory reactivation was not significantly correlated with memory integration measured in the Direct test ($r_s$ (125) = -0.02, $p$ = .803) or relative model fit in the Transfer test ($r_s$ (104) = 0.06, $p$ = .543). Furthermore, temporal delay did not modulate the strength of memory reactivation ($F$(2, 122) = 0.11, $p$ = .894; main effect $BF_{01}$ = 1240.96; 25 < pairwise $BF_{01}$ < 35) or the correlation between memory reactivation and memory integration ($F$(2, 116) = 1.47, $p$ = .234; main effect $BF_{01}$ = 7.24; 1 < pairwise $BF_{01}$ < 5). Thus, while we have evidence for memory reactivation, we did not find evidence for it specifically supporting memory integration at long delays. Instead, the correlation between memory reactivation and memory integration at the longest delay was negative ($r_s$ (36) = -0.33, $p$ = .044).

### Might perseverating on Session 1 locations during Session 2 learning influence representations?

Lastly, we asked if the extent to which participants perseverate on the Session 1 reward locations after exposure to Session 2 rewards might relate to the structure of their memories. Conceivably, this continued reactivation could encourage either integration (by highlighting similarities) or differentiation (by highlighting differences). To assess these alternate possibilities in an exploratory analysis, we operationalized perseveration as the difference in time spent in the Alternate compared to the Adjacent zone ($TIZ_{Alt}$-$TIZ_{Adj}$) during Session 2 no-feedback trials. In this metric, positive values reflect more time spent in the previously rewarded zone than a never-rewarded one equidistant from the current Target (i.e., greater perseveration).

As expected, participants perseverated on Session 1 locations less as they gained more Session 2 experience across trials ($F(5.34, 683.34) = 2.70$, $p = 0.018$; see Fig S9 in S2 File). While there was no significant effect of perseveration across the entire session ($\beta = 0.01$, 95% CI = [0, 0.02]), we did find reliable perseveration in the first no-feedback trial ($\beta = 0.04$, 95% CI = [0.01, 0.06], Fig 5A)—consistent with the idea that perseveration occurred but was very short-lived. Accordingly, we used the first no-feedback trial to index perseveration below; it did not vary significantly as a function of delay condition ($F(2, 126) = 1.89$, $p = 0.156$; main effect $BF_{01}$ = 377.51; 6 < pairwise $BF_{01}$ < 32).

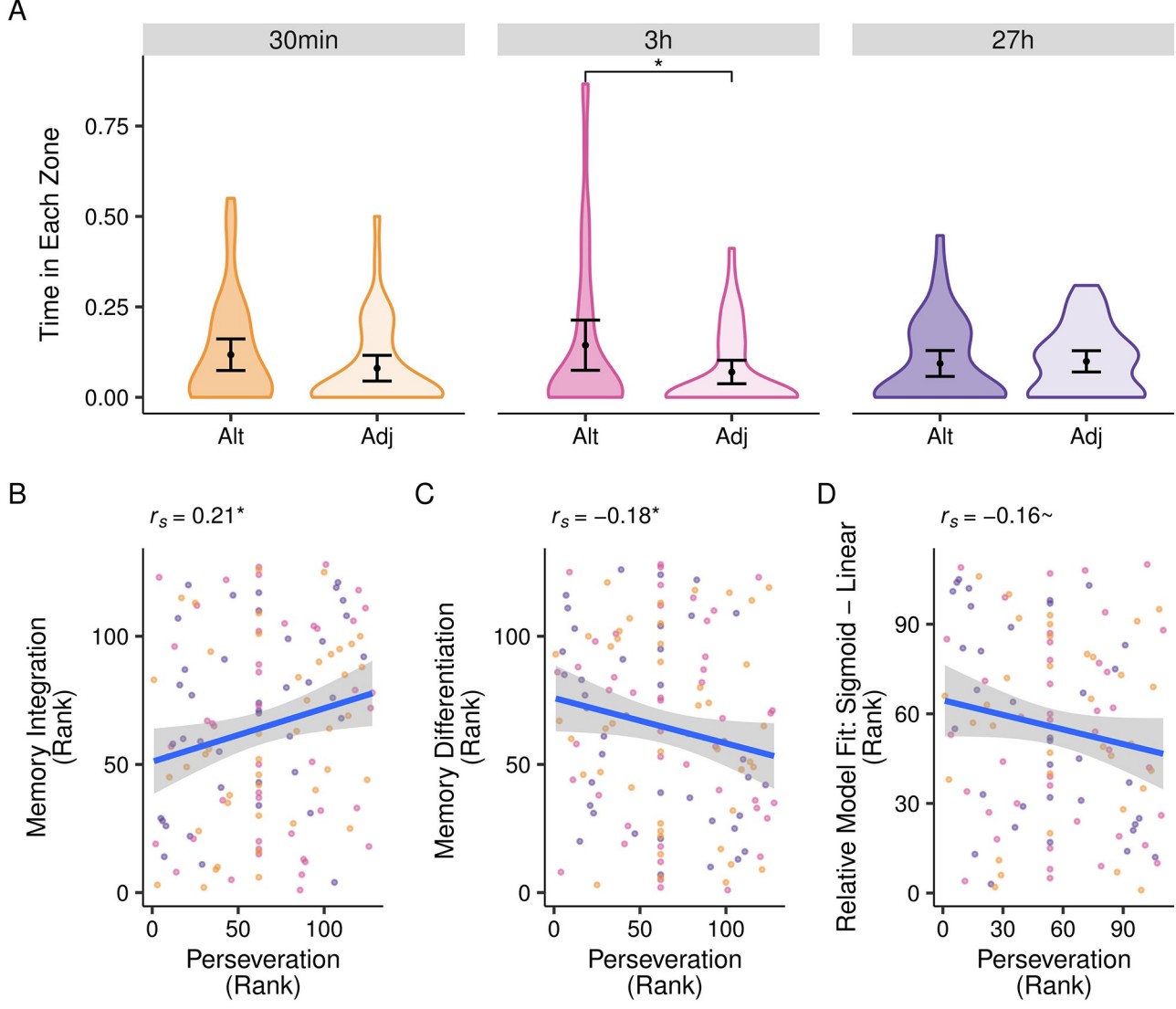

**Fig 5. Perseveration on Session 1 locations during Session 2 learning.** (**A**) Participants generally spent more time in the Alternate as compared with the Adjacent zone during the first trial of Session 2 learning (collapsed across delay conditions; not depicted), our index of perseveration. Individual differences in perseveration were significantly related to (**B**) memory integration and (**C**) memory differentiation from the Direct test, but not (**D**) relative fit of sigmoid vs. linear models from the Transfer test. For panels B-D, many participants tied in perseveration rank because they spent time in neither the Alternate nor Adjacent zone during the first trial. Results remained similar when these participants were excluded (data not reported). *$p<0.05$, ~$p<0.10$.

Investigating individual differences in perseveration on the first no-feedback trial of Session 2 learning revealed an interesting dissociation: Perseveration was associated positively with integration ($r_s$ (126) = 0.21, $p$ = .019; Fig 5B) and negatively with differentiation ($r_s$ (126) = -0.18, $p$ = .047; Fig 5C) on the Direct test (the correlation with attractor states from Transfer test only trended toward significance, though in the expected direction; $r_s$ (108) = -0.16, $p$ = .091; Fig 5D). Importantly, these associations were not present when we considered perseveration for the last rather than the first no-feedback trial in Session 2 (integration: $r_s$ (126) = 0.11, $p$ = .231; differentiation: $r_s$ (126) = -0.06, $p$ = .499), suggesting that the observed link between early perseveration and subsequent integration is not due simply to participants' resistance to learning the new locations. Moreover, the relationships of early perseveration with both integration and differentiation did not vary by delay (integration: $F(2, 120)$ = 0.64, $p$ = .531; main effect $BF_{01}$ = 14.39; 2 < pairwise $BF_{01}$ < 5; differentiation: $F(2, 120)$ = 0.37, $p$ = .693; main effect $BF_{01}$ = 17.43; 2 < pairwise $BF_{01}$ < 5). Together, these results suggest that leveraging Session 1 memories early during Session 2 learning may serve to highlight the similarities between experiences and foster integration—and do so regardless of the delay between experiences.

## Discussion

Here, we drew on compelling demonstrations that temporal proximity powerfully shapes how memories are integrated in the mouse brain, with experiences that occur close in time (6h or less) linked in memory and those learned across longer intervals (18h or more) stored independently. We ask whether time analogously influences how humans integrate or differentiate their learning in related spatial environments. To this end, we validated a new appetitive spatial memory paradigm by showing that estimates of memory differentiation were correlated across the final Direct and Transfer tests and that integration and differentiation correlated with the degree to which individuals expressed Session 1 learning during Session 2. However, our predictions about temporal delay did not bear out in the data: Participants showed evidence of simultaneous integration and differentiation, but critically this effect did not differ across short (30min and 3h) or long (27h) delay conditions in the vast majority of our measures. We observed a significant effect of delay in a follow-up analysis run on a subsample of participants who had particularly good memory in the final test. In all other (planned) analyses, however, we saw Bayesian evidence for null effects of delay at "strong" to "decisive" levels. Thus, we identified important limits on time's power to shape memory representations—limits that must be reconciled with existing demonstrations of temporal proximity effects in both rodent and human memory.

One possible explanation lies in our decision to limit our long delay interval to only 27h. There were many good reasons for this decision. Co-allocation experiments in rodents have consistently found that memories formed at least 18-24h apart are not integrated in the same way as those formed less than 6h apart [1, 2, 4], and that this time window is related to changes in CREB [4] and CCR5 [1] that occur as early as 12h after the initial learning. Indeed, a recent computational model of the phenomenon simulated neural excitability as being elevated 12h after initial learning [44] based on this work. Though some studies have used a longer delay of 7 days [3, 32], choosing such a long window opens the door to memory differentiation being related to hippocampal neurogenesis, which has also been hypothesized to separate memories encoded at different times [45–48]. Thus, our choice of delays is well grounded in the most pertinent literature.

However, research related to these memory co-allocation studies has shown that neurons can retain their heightened excitability for 1–7 days after learning [15, 16, 49–51], suggesting that longer delay manipulations may be necessary in some contexts. While the factors

determining excitability duration need more exploration, experiments with longer windows tend to use extensive training, such as delivering multiple shocks [16, 49] or hours of eye-blink conditioning [15, 51]. Indeed, the magnitude of heightened excitability can scale with the extent of training [51]. Thus, it is possible that our extensive training procedure resulted in a longer window for memory integration than what has been seen in related human [31] and rodent [1, 2, 4] work, in which each "memory" was acquired within a few seconds or minutes. Importantly, though, extending the delay manipulation beyond 27h to test this possibility would require a major revision to our paradigm. Specifically, we iteratively designed our protocol to ensure that there was minimal forgetting of Session 1 locations at the 27h delay, as forgetting the first memory could result in performance that looks like integration (i.e., only searching the more recent reward locations) simply because only one is available—a possibility not thoroughly addressed in related work using very long delays [3, 32]. Indeed, we may have unmasked a hint of the predicted delay effect when only considering participants who could remember both reward locations during the final test, namely those that would not have been misclassified as having an integrated memory due to forgetting. These results, combined with the moderate final test performance levels and strong recency bias, suggest that participants were not overtrained and that a longer delay condition would not have been feasible. Accordingly, future work using alternative paradigms is required to assess how the strength of learning influences the duration of integration windows.

Alternatively, perhaps the nature of the spatial learning elicited by our task explains the unanticipated results. Specifically, we taught participants about distributions of rewarded locations to allow for spatial memory integration—distributions that may themselves result in integrated schematic knowledge, even within a single session. Accordingly, participants may have relied on schematized representations of these distributions, supported by neocortical regions [52–56]. It was recently shown that memory co-allocation does not occur elsewhere in the cortex in the same way as it does within MTL structures [2]. So, time may not influence the integration of more schematic knowledge in memory in the same way that it does single events. We, however, reasoned that in our task, memory for reward locations would remain hippocampally-dependent across the delay because mice performing a similar task only extracted the underlying distribution and exhibited neocortex-dependent memories—both possible evidence of schematization—at 28 days but not 24 hours after acquisition [56] (see also [57–59]). A recent study, though, points to some species differences: In a task modeled off [56], humans appeared to extract schematic knowledge of distributions when tested at delays as short as 15 minutes [60]. Thus, the nature of our task may likewise yield interesting species differences in how time influences the integration of memories across sessions—differences which could be assessed by using this paradigm in rodents.

Beyond the structure of the task, an alternative possibility is that the shared context in which the task was performed may have bridged memories, regardless of the time separating them. Specifically, due to the COVID-19 pandemic, we tested participants while they were in a quiet spot in their homes. All sessions were conducted over Zoom to ensure that the location was quiet and free of distractors, but experimenters otherwise had no control over the testing environment. If participants chose the same location across sessions, this shared context could have facilitated the retrieval of Session 1 locations during Session 2 learning, driving their integration even at long delays. Indeed, both retrieval prior to related learning [4] and co-retrieval [59] have been shown to increase memory co-allocation in the rodent MTL, consistent with longstanding theories of integrative encoding in humans [61–63]. In our data, we see evidence for Session 1 reactivation and perseveration prior to and early on in Session 2 learning, respectively, suggesting that the shared spatial context across sessions—or perhaps the context of the experiment itself—may have bridged memories across delays. Indeed, continuing to search in

the original Session 1 locations early during Session 2 was positively associated with integration and negatively associated with differentiation, suggesting that continued reactivation might have highlighted similarities and facilitated the merging of representations. However, this perseveration effect did not differ across our delay conditions, nor did it specifically relate to integration across long delays. On the other hand, another human study [32] underscored the importance of context by showing that delay only influenced memory performance when each session (including the test session) was performed in a distinct, different room. However, the contextual conditions under which the effect of delay was observed were quite inconsistent across measures, with the most delay-sensitive physiological measures only showing it when participants were tested in one of the original learning contexts—that is, under the exact conditions in which it was not seen in their memory performance. Further, research in rodents has shown that time influences hippocampal neuron firing despite spatial context manipulations [26, 64]. Thus, whether and how spatial context comes to overwhelm the influence of time on memory in humans—but not rodents—remains unknown.

All this reasoning is centered on the type of integrative encoding mechanisms identified in the related rodent studies [3, 4]—in which related experiences are encoded in overlapping sets of neurons—but the performance of our human participants may have been influenced by multiple integrative and interference processes. Indeed, developing clear operational definitions of integration—and generalization more broadly—has been a theoretical thorn in the side of our field [65] because multiple mechanisms dictate how memories interact at both the time of encoding and retrieval. Broadly construed, behaviours reflecting memory integration could be supported by three encoding processes [37]: (1) fusing experiences within one representation, potentially at the cost of idiosyncratic experiential details [66, 67], (2) explicitly linking two separately represented experiences [68, 69], for example, by building a complex situation model [70], and (3) separately encoding memories that can later be recombined at the time of retrieval [71, 72]. With the goal of targeting the first mechanism (i.e., the one most relevant to the neurobiological impact of delay), we used a task and behavioural indices that did not distinguish between the generalization and interference that can result from a fused representation. However, other types of integration and/or interference could contribute to task performance. For example, participants could receive high scores on our integration index by storing separate memories of each session that then compete with each other at the time of retrieval. Indeed, this type of retrieval competition has been shown to underlie many demonstrations of memory interference [73–75]. Notably, a recent fMRI study showed that this kind of retrieval competition depends on how memories are reactivated and integrated during related learning [76]. Thus, while our focus has been on integrative encoding mechanisms, the role of interconnected interference during retrieval should not be discounted.

Relatedly, delay's null impact on memory integration and differentiation may reflect limitations in our ability to infer these memory representations from behaviour. In our Direct test, we defined differentiation as spending more time in the Target than Alternate zone, and integration as spending more time in the Alternate than Adjacent zone. Therefore, memory precision could contribute to our differentiation scores, as differentiation requires participants to both remember the reward locations of different arenas and to do so with enough precision to not stray into the Alternate zone. By contrast, integration would not be influenced by precision. There are several reasons to believe that this imbalance did not overly influence our primary results, though. First, the delay groups did not differ in their expression of memory precision in our independent virtual navigation task, suggesting similar spatial memory precision across conditions. Therefore, it is unlikely that individual differences in memory precision would have impeded our ability to detect true delay-dependent differences in integration or differentiation. Second, we also statistically controlled for individual differences in spatial

memory ability by using Session 1 learning (which did show modest delay differences) as a covariate in all analyses reported in the main text. Third, we purposefully used large zone definitions, encompassing nearly a full quadrant of the arena. Accordingly, even imprecise memories would be sufficient to get participants to the target zone. Lastly, this differentiation index was strongly correlated with sigmoidal fits in the Transfer test (which would be less directly influenced by precision), suggesting that they track the same underlying construct.

Lastly, the motivational state should be considered as a potential factor driving our null results. Our experiment is unique among those studying temporal proximity and memory integration in that it used an appetitive learning paradigm rather than fear conditioning [2–4, 32], or unmotivated learning [31]. This design decision filled a crucial gap for understanding how time influences memory integration across motivational states. And—while speculative— it may very well be that motivational states act as boundary conditions on the phenomenon. In particular, appetitive tasks are more likely to engender behavioural activation [77] or interrogative states [78], which encourage the integration of disparate elements both within and across experiences. By contrast, immediate threats are more likely to engender an imperative state [78] in which participants would be stuck in the here and now, reducing reactivation of all but recent experiences. This could explain why we see some level of integration across all delay conditions, while fear-motivated tasks only show integration at short delays. Notably, arousal should determine how the valence of a task translates into these motivational states [77, 78]. Specifically, appetitive tasks can also engender imperative states when the stakes (and arousal) are high. The distribution of small rewards across our long (and admittedly monotonous) training sessions, thus, likely biased people to take on an interrogative, integrative motivational state. Notably, though, if rodents were to perform our task, the arousal conditions could be quite different, with the testing session and opportunity for reward being the most arousing aspect of their day. Thus, even within the same task, the differential adoption of motivational orientations across species could drive distinct memory representations.

Beyond assessing temporal proximity's influence on spatial memory representations, our novel paradigm made important additions to the understanding of how related environments are represented in human memory. The existing literature suggests that spatial memories tend to be highly differentiated [34, 35, 79], perhaps owing to their dependence on hippocampal networks optimized to separate [80, 81] and differentiate [82] similar experiences. However, separately estimating the differentiation and integration of the two learned environments in our task revealed a mixture of both types of representations. Thus, the differentiation of similar environments is not obligatory, and rather may be related to people's goals while exploring them. For example, our task was carefully designed such that integrating memories could be helpful, on one hand, because reward distributions were located in close proximity to one another with respect to the global environment. But on the other, differentiation could be helpful because the distributions did not overlap. By contrast, tasks that have elicited differentiated representations required participants to navigate to different locations depending on subtle differences across environments [34, 35, 79]. Notably, one study in which navigation paths could sometimes be transferred between virtual cities also found a mixture of differentiated and merged representations in both participants' brains and behaviours [83]. Alternatively, the mixture of integration and differentiation behaviour seen in our study may reflect multiple neural representations contained within the network comprising the memory engram. For example, anterior and posterior hippocampal regions have been shown to simultaneously represent related experiences in integrated and differentiated ways [84]. Notably, artificially silencing co-allocated engram neurons in the MTL can unmask the constituent distinct memories (stored elsewhere) by removing their link [2]. These findings serve as an important reminder of the complex link between behaviour and distributed memory representations.

To conclude, here we demonstrated that the temporal proximity of experiences does not always influence their integration in human memory. Our largely null findings are in sharp contrast to predictions drawn from the striking impact of temporal proximity on rodent memory representation, and two (somewhat more nuanced) demonstrations of analogous effects in human memory for images [37] and their associations [32]. In our dissection of the present surprising findings, we raise many underlying factors that could limit temporal proximity's impact on memory representation. We hope that raising these possibilities—and developing an important new paradigm that could test some of them—will inspire future research to answer key questions about how time shapes memory across species.

## Supporting information

**S1 File.**
(PDF)

**S2 File.**
(PDF)

**S3 File.**
(PDF)

## Acknowledgments

The authors would like to thank the following individuals for their dedication to protocol testing and data collection: Zainah Azam, Stella Mo, Sarah Eisen, Rekha Ravikumar, Angel Chang, Mila Valcic, Elisabetta Canaletti, Yihan Qiu, Vanneza Moosa, Dorothy Sun, Sinem Kocamanoglu, Padideh Hassanpour, Jacqualine Yeh, Magda Binczyk. The authors would also like to thank Sarah Berger, Zahra Abolghasem, and Emily Wang for their help with study coordination, and Yimeng Zhang for helpful discussions on statistical analysis.

## Author Contributions

**Conceptualization:** Xiaoping Fang, Benjamin Alsbury-Nealy, Paul W. Frankland, Sheena A. Josselyn, Margaret L. Schlichting, Katherine D. Duncan.

**Data curation:** Xiaoping Fang.

**Formal analysis:** Xiaoping Fang, Benjamin Alsbury-Nealy.

**Funding acquisition:** Paul W. Frankland, Sheena A. Josselyn, Margaret L. Schlichting, Katherine D. Duncan.

**Methodology:** Xiaoping Fang, Benjamin Alsbury-Nealy, Ying Wang, Margaret L. Schlichting, Katherine D. Duncan.

**Project administration:** Xiaoping Fang, Benjamin Alsbury-Nealy, Margaret L. Schlichting, Katherine D. Duncan.

**Software:** Benjamin Alsbury-Nealy.

**Supervision:** Margaret L. Schlichting, Katherine D. Duncan.

**Visualization:** Xiaoping Fang.

**Writing – original draft:** Xiaoping Fang, Margaret L. Schlichting, Katherine D. Duncan.

**Writing – review & editing:** Xiaoping Fang, Benjamin Alsbury-Nealy, Ying Wang, Paul W. Frankland, Sheena A. Josselyn, Margaret L. Schlichting, Katherine D. Duncan.

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
