## [Decision Letter · Decision Letter 0]

15 Feb 2023

PONE-D-22-35255Time separating spatial memories does not influence their integration in humansPLOS ONE

Dear Dr. Fang,

Thank you for submitting your manuscript to PLOS ONE. After careful consideration, we feel that it has merit but does not fully meet PLOS ONE’s publication criteria as it currently stands. Therefore, we invite you to submit a revised version of the manuscript that addresses the points raised during the review process.

The revised version of the manuscript should address all comments raised by the two reviewers. Please pay particular attention to Reviewer 1's comments with respect to the lack of public availability of the data and the broken link for the pre-registration of wave 2. I experienced the same issues. 

We look forward to receiving your revised manuscript.

Kind regards,

Bradley R. King

Academic Editor

PLOS ONE

Journal Requirements:

2. Please note that in order to use the direct billing option the corresponding author must be affiliated with the chosen institute. Please either amend your manuscript to change the affiliation or corresponding author, or email us at plosone@plos.org with a request to remove this option.

Reviewers' comments:

Reviewer's Responses to Questions

**Comments to the Author**

1. Is the manuscript technically sound, and do the data support the conclusions?

Reviewer #1: Yes

Reviewer #2: Yes

2. Has the statistical analysis been performed appropriately and rigorously? 

Reviewer #1: Yes

Reviewer #2: Yes

3. Have the authors made all data underlying the findings in their manuscript fully available?

Reviewer #1: No

Reviewer #2: Yes

4. Is the manuscript presented in an intelligible fashion and written in standard English?

Reviewer #1: Yes

Reviewer #2: Yes

5. Review Comments to the Author

Reviewer #1: Note: For the question above, "Have the authors made all data underlying the findings in their manuscript fully available?" I said "no" because the OSF project they link to requires me to request access before I can view it. But the dataset does appear to have been deposited, though I can't actually see it and will not request access in order to avoid breaking reviewer anonymity.

Summary

The authors introduce a paradigm designed to bridge between human and rodent work examining how memories acquired in different environments are stored, and how integration and differentiation of environments in memory affects behavior. In contrast to rodent work that has generally focused on fear conditioning, the authors use an appetitive learning paradigm where participants learn a set of locations that are rewarded upon navigating to them. Participants learned about reward locations in two arenas with the same global cues but a different shape of enclosure. The authors focus on two main measures of behavior in unrewarded trials: (1) discrimination, the time spent visiting the rewarded locations specific to the current arena compared to locations that were rewarded in the other arena, and (2) integration, the time spent visiting rewarded locations specific to the other arena, relative to a location that is equidistant from the rewarded locations of the current arena. The authors also examine reward-seeking behavior in novel arenas of different shapes, to measure generalization of behavior to new environments. They find that participants with more evidence of discrimination in the main environments also show a predicted pattern of results consistent with behavior being guided by one environment or the other, but not both. In contrast to their main predictions, the authors find that the time delay between environments (30 minutes, 3 h, or 27 h) does not affect memory integration, as assessed by their environment confusion measures. They conclude that there are boundary conditions on the effect of delay on memory integration, which may relate to paradigm or species differences compared to previous work.

Evaluation

The authors address an important topic of temporal delay effects on memory representation that have been raised by recent research. The manuscript makes good use of study design and pre-registered statistical analyses. The main finding of a null effect of temporal delay on memory integration appears to be well established. Many questions remain about what the specific boundary conditions are on temporal delay effects, but the manuscript provides a valuable contribution in exploring the effect under different conditions.

There is a focus throughout on very general mechanisms of integration and differentiation that I believe should be qualified to reflect the assumptions underlying those theoretical conclusions. In particular, effects that are here called "integration" have previously been labeled as reflecting "interference", and may reflect retrieval rather than purely reflecting memory storage, as "integration" is often understood. The terminology used should be clarified, and more discussion is necessary to make clear what behavioral effects have been shown here compared to what theoretical conclusions are being inferred from the results. As discussed below, some results are difficult to interpret, and may have alternate interpretations that should be considered.

Major issues

1. What is meant by "integration" is somewhat unclear. Memory integration, wherein two memories are represented similarly to one another, has previously been proposed as an important factor in determining behavior. As noted in the introduction, there is evidence that memory integration, at the level of individual engrams, is a factor in determining fear expression in rodents. Here, however, there is no measurement of memory engrams; instead, the manuscript reports evidence of interference effects, which are interpreted as evidence for integration. Depending on your working definition of "integration", this may be reasonable, but integration seems likely to be taken by some readers to mean something very specific about memories for the two environments being coded by overlapping engrams. However, that need not be the case here; global cues could trigger retrieval of either environment, causing interference, without there needing to be any overlap between engrams. It's reasonable to discuss potential implications of the results for memory integration (particularly given the main predictions of the effect of delay on engram integration), but it's also important to be clear when discussing the results that interference effects may also arise from other mechanisms without there needing to be engram overlap.

2. The main measure of differentiation is ambiguous, as it may reflect at least two different sources of individual variability: (1) interference between environments, potentially due to memory integration (as hypothesized in the manuscript), and (2) spatial precision of memories within a given environment. If spatial precision is higher for a given participant, such that they consistently respond close to the rewarded area and not further from the rewarded area, then the TIZtarget-TIZalternate difference will tend to be higher compared to participants with lower spatial precision. In that case, some of the variability in individual differences would reflect variability in spatial precision rather than differentiation between the two environments. The integration measure does not suffer from this issue, as both zones of interest there are equidistant from the target zone.

The strong correlation with the fit of a sigmoidal model reduces this concern, as it's reasonable to think there is a theoretical link between those measures, but still, memory precision (in terms of specific spatial location) could also be correlated with attractor-like dynamics across environment morphs, in addition to memory differentiation (in terms of whether the environments are confused). The (likely small) potential effect of spatial precision on the differentiation measure should be discussed. Optionally, there might be some way to control for it by measuring the spatial precision of responses and controlling for variance in spatial precision when measuring individual differences in differentiation.

3. The higher accuracy on no-feedback trials in the second environment is interpreted as potentially being evidence for integration, i.e., that previous experience with the first environment leads to improved learning. However, that finding also seems compatible with participants improving at the task through other means, such as learning how to orient using wall landmarks, or getting better at seeking reward efficiently. To show clearer evidence of a transfer effect, ideally you would compare performance on a second environment that shares global cues with the first to performance in a new environment that does not share those cues, and show that there is improved performance only when global cues are shared with the first environment. The alternative interpretation of environment performance differences, i.e., that performance may be affected by general task learning rather than necessarily reflecting environment transfer, should be discussed.

4. It's unclear that a linear model fit to performance across the different environment morphs would necessarily indicate integration. The curves reported in the supplement that best fit a linear model tend to be flat, indicating that the balance of circle and square reward region searching did not change between the different morphs. That pattern is potentially consistent with integration, especially for participants with no bias toward one reward region or the other, but many participants seem to be biased to look for reward near one environment's target region or the other. This seems consistent with them simply remembering one environment and responding in accordance with that, rather than necessarily retrieving an integrated representation of both environments. The inclusion criteria for this analysis only required above-chance memory for at least one of the two arenas (page 18), which allows for the possibility that included participants only remembered one of the arenas.

Minor issues

The deposited data on OSF currently cannot be accessed without requesting access from the authors.

Page 13: The preregistration for wave 2 (https://osf.io/gnxpt) is currently not publicly accessible.

Page 14: More detail should be given about how the study was run, especially given the online nature of the study. How was the study run on participants' computers? For example, was the study run using a web browser? If so, were there any requirements for which browser participants must use, or requirements for the computer? Are there any details of how you monitored performance remotely that might be useful for other researchers conducting online studies?

Page 15: Please give some more detail about the norming data. How many participants were in the norming study? How did they rate perceptual discriminability? Were they shown first-person images of the environments, or maps, or did they navigate the environments?

Page 15: More detail should be specified for the environment familiarization phase. How was location displayed on the aerial view? From Figure S1, it looks like a triangle was shown, but the screen cap is small and hard to see; in any case, this would be good to mention in the methods text also. Was starting location randomized? If so, were there any constraints on the randomized placement?

Figure 4: "Cmpt" (short for competing?) should be "Alt".

What was the delay between waves 1 and 2, and when were they collected? This might be relevant especially if data were collected during the early stages of the pandemic, when conditions were changing rapidly and a delay between data collection of different conditions could be a concern. Of course, the various control analyses you included to test for cohort differences would reduce any such concern.

Optional suggestions

Memory integration could lead to intrusion of the non-current reward locations, as predicted in the manuscript, but I wonder if there could also be a more subtle bias introduced by memory integration. If memories for the environments are truly merged, could this lead to participants searching for reward in a "compromise" area between the two locations in the different environments? Looking for this could involve testing to see if responses in one environment are closer to the alternate zone compared to the adjacent zone, to see if there is a subtle nudge in the direction of the alternate reward locations, even when the responses are not in the alternate zone itself.

Reviewer #2: This manuscript details a well-designed and interesting study examining how temporal proximity influences the integration or differentiation of memories. The task is designed to translate paradigms from rodent research to humans, a commendable and important endeavor, and the results add to the growing literature understanding the boundaries on encoding processes that lead to integration/differentiation. I do have several questions and comments, as detailed below:

1. Clarification on methods/terminology. How overlapping were the 6 ‘points’ pulled from the gaussian distribution? Were participants instructed that the rewards would not always be in the same location on each exploration opportunity? Were the quadrants of rewards in the circle and square arena always in the same relation to one another (e.g., in Figure 1A, they appear to be in adjacent quadrants, raising the possibility that it may be easier to learn the second arena if the reward distribution center was closer to the first)

2. Were there any order effects in learning between the two arenas (e.g., was it easier to learn the circle before the square, or vice versa?)

3. Was the extent to which participants showed a recency bias for session 2’s location on the direct test related to how quickly participants acquired the session 2 location or generally how quickly participants learned the point locations during reward search? I’m wondering if there are individual differences that could predict and unpack this bias more.

4. Somewhat related to question 3, the authors could also examine the speed of learning for the second arena compared to the first, rather than just the integration behavior on the direct test. Does the extent to which participants continue to perseverate on the learned reward location in the first arena have any relationship with later integration (or differentiation) on the direct test? In addition to the authors hypothesis that reactivation could facilitate ‘bridging’ across memories and thus support integration, it could also be the case that the longer participants continue to search in the prior location without finding a reward could serve to highlight differences across the arenas. It would be interesting to see how learning ability may mediate the relationship between reactivation and integration or differentiation. The authors do note generally that the 3hr group was faster on both arenas, but what about the other groups?

5. I found the claim in the first paragraph of the results section (on the initial learning performance), that the findings of stronger learning by the end of session 2 constitute a transfer effect attributed to memory integration in rodent work to be somewhat unclear – are the authors suggesting this pattern has been shown in studies that show shared neural populations? Otherwise, why wouldn’t this simply be a function of task practice? Perhaps consider operationalizing the term or explaining the findings on the transfer effect more thoroughly in the introduction.

6. Why include the two arenas again during the transfer test? Couldn’t this re-exposure serve as anchors that may bias behavior on the morphs, especially considering that they were also just tested on the direct test?

6. PLOS authors have the option to publish the peer review history of their article (what does this mean?). If published, this will include your full peer review and any attached files.

Reviewer #1: No

Reviewer #2: No

---

## [Author Response · Author response to Decision Letter 0]

4 Apr 2023

Response to reviews has been uploaded as an independent file.

---

## [Decision Letter · Decision Letter 1]

26 Apr 2023

PONE-D-22-35255R1Time separating spatial memories does not influence their integration in humansPLOS ONE

Dear Dr. Fang,

Thank you for submitting your manuscript to PLOS ONE. After careful consideration, we feel that it has merit but does not fully meet PLOS ONE’s publication criteria as it currently stands. Therefore, we invite you to submit a revised version of the manuscript that addresses the points raised during the review process. Please see specific comments from Reviewer 1 below. There is a missing pre-registration link on one page. More importantly, the data points themselves are missing from a series of scatter plots. 

We look forward to receiving your revised manuscript.

Kind regards,

Bradley R. King

Academic Editor

PLOS ONE

Journal Requirements:

Reviewers' comments:

Reviewer's Responses to Questions

**Comments to the Author**

1. If the authors have adequately addressed your comments raised in a previous round of review and you feel that this manuscript is now acceptable for publication, you may indicate that here to bypass the “Comments to the Author” section, enter your conflict of interest statement in the “Confidential to Editor” section, and submit your "Accept" recommendation.

Reviewer #1: (No Response)

Reviewer #2: All comments have been addressed

2. Is the manuscript technically sound, and do the data support the conclusions?

Reviewer #1: Yes

Reviewer #2: Yes

3. Has the statistical analysis been performed appropriately and rigorously? 

Reviewer #1: Yes

Reviewer #2: Yes

4. Have the authors made all data underlying the findings in their manuscript fully available?

Reviewer #1: Yes

Reviewer #2: Yes

5. Is the manuscript presented in an intelligible fashion and written in standard English?

Reviewer #1: Yes

Reviewer #2: Yes

6. Review Comments to the Author

Reviewer #1: My major concerns have been addressed by the manuscript revisions, through new analyses and discussion points. I note a few remaining minor concerns below.

Minor issues:

Page 6: The two preregistration links are the same. The link for wave 2 is shown twice, and the preregistration for wave 1 (https://osf.io/49dtx) is not shown here.

Figure 3 - In my copy of the PDF, there are no points shown for the scatterplots in panels B and C, only a regression line. These points were previously visible in the original version and appear to not have been modified in the revision, so this was not an issue for evaluating the revision.

Figure 5 - I also see no scatterplot points in my copy of the PDF for panels B, C, and D.

Figure S4 - I believe "Cmpt" should be "Alt".

Figure S7 - In my copy, I don't see any points in panels A, B, C, and D. These points were shown in the original manuscript, and the results seem to not have changed in the revision, so this was no an issue for evaluating the revision.

Reviewer #2: (No Response)

7. PLOS authors have the option to publish the peer review history of their article (what does this mean?). If published, this will include your full peer review and any attached files.

Reviewer #1: No

Reviewer #2: No

---

## [Author Response · Author response to Decision Letter 1]

10 May 2023

response has been uploaded as a separate file

---

## [Editor Report · Decision Letter 2]

15 May 2023

PONE-D-22-35255R2Time separating spatial memories does not influence their integration in humans

PLOS ONE

Dear Dr. Fang,

Thank you for submitting your manuscript to PLOS ONE. Upon review of the updated figures, there are additional issues that warrant attention. Please see a detailed list below. We invite you to submit a revised version of the manuscript that addresses the points raised during the review process.

We look forward to receiving your revised manuscript.

Kind regards,

Bradley R. King

Academic Editor

PLOS ONE

Journal Requirements:

Additional Editor Comments:Figure 3B – In the last round of review, one reviewer noted that the data points in the scatter plots were not visible. In the revised R2 version, the data points are now present. However, the data do not match what was submitted in the original submission (R0). At first glance, it appeared that the color coding simply changed, but this is not the only change/difference (although it appears to be one change). Most noticeable difference with respect to the data is the pink data point in the original version that is at coordinates ~(75, 5) that is no longer present in the revised version. In the revised version, **please explain why this image is no longer the same as well as the impact on the reported results. **Figure 5, S8 and S9 - these were all added as part of R1 (not included in R0). In light of the discrepancy noted above for Figure 3, **please confirm that these images and the corresponding statistics are accurate. **Figure S7; Panel E – When reviewing the updated figures, I noticed that the color coding changed from the original version to R1 (remained the same from R1 to R2). Specifically, data originally labeled in R0 as the 30m group were re-labeled as 27h in R1, data originally labeled as 3h were re-labeled as 30mn in R1, and data originally labeled as 27h were re-labeled as 3h in R1. I didn’t see this change explicitly mentioned in the previous response document.  **Given the lack of a significant effect, it appears this change had no impact on the results. Can you confirm? And please also confirm that the most recent version of the figure (in R1 and R2) is indeed the correct one. **Figure S7 Panels A-D and Figure 3B – The datapoints are now visible and match what was in the original version with no change to the corresponding results. Accordingly, issues surrounding these images are considered addressed.

---

## [Author Response · Author response to Decision Letter 2]

11 Jul 2023

Responses to reviewers have been uploaded as a separate file.

---

## [Editor Report · Decision Letter 3]

24 Jul 2023

Time separating spatial memories does not influence their integration in humans

PONE-D-22-35255R3

Dear Dr. Fang,

I thank you for taking the time to thoroughly revisit all analyses / figures and for making all data and scripts publicly available.

We’re pleased to inform you that your manuscript has been judged scientifically suitable for publication and will be formally accepted for publication once it meets all outstanding technical requirements.

Kind regards,

Bradley R. King

Academic Editor

PLOS ONE
---

## [Editor Report · Acceptance letter]

31 Jul 2023

PONE-D-22-35255R3 

Time separating spatial memories does not influence their integration in humans 

Dear Dr. Duncan:

I'm pleased to inform you that your manuscript has been deemed suitable for publication in PLOS ONE. Congratulations! Your manuscript is now with our production department. 

Kind regards, 

on behalf of

Dr. Bradley R. King 

Academic Editor

PLOS ONE